# Morphological and Mechanical Properties of Bilayers Wood-Plastic Composites and Foams Obtained by Rotational Molding

**DOI:** 10.3390/polym12030503

**Published:** 2020-02-25

**Authors:** Roberto Carlos Vázquez Fletes, Erick Omar Cisneros López, Francisco Javier Moscoso Sánchez, Eduardo Mendizábal, Rubén González Núñez, Denis Rodrigue, Pedro Ortega Gudiño

**Affiliations:** 1Department of Chemical Engineering and CERMA, Université Laval, Quebec, QC G1V 0A6, Canada; Denis.Rodrigue@gch.ulaval.ca; 2Departamento de Ingeniería Química, CUCEI, Universidad de Guadalajara, Blvd. Gral. Marcelino García Barragán 1421, Guadalajara, Jalisco 44430, Mexico; ruben.gnunez@academicos.udg.mx; 3Departamento de Física, CUCEI, Universidad de Guadalajara, Blvd. Gral. Marcelino García Barragán 1421, Guadalajara, Jalisco 44430, Mexico; erick.cisneros@academicos.udg.mx; 4Departamento de Química, CUCEI, Universidad de Guadalajara, Blvd. Gral. Marcelino García Barragán 1421, Guadalajara, Jalisco 44430, Mexico; francisco.moscoso@academicos.udg.mx (F.J.M.S.); eduardo.mmijares@academicos.udg.mx (E.M.)

**Keywords:** natural fibers, foams, bilayer materials, rotational molding, thermal properties, morphology, mechanical properties

## Abstract

In this work, the suitability for the production of sustainable and lightweight materials with specific mechanical properties and potentially lower costs was studied. Agave fiber (AF), an agro-industrial waste, was used as a reinforcement and azodicarbonamide (ACA) as a chemical blowing agent (CBA) in the production of bilayer materials via rotational molding. The external layer was a composite of linear medium density polyethylene (LMDPE) with different AF contents (0–15 wt %), while the internal layer was foamed LMDPE (using 0–0.75 wt % ACA). The samples were characterized in terms of thermal, morphological and mechanical properties to obtain a complete understanding of the structure-properties relationships. Increases in the thicknesses of the parts (up to 127%) and a bulk density reduction were obtained by using ACA (0.75 wt %) and AF (15 wt %). Further, the addition of AF increased the tensile (23%) and flexural (29%) moduli compared to the neat LMDPE, but when ACA was used, lower values (75% and 56% for the tensile and flexural moduli, respectively) were obtained. Based on these results, a balance between mechanical properties and lightweight can be achieved by selecting the AF and ACA contents, as well as the performance and aesthetics properties of the rotomolded parts.

## 1. Introduction

In recent times, conventional plastics have become indispensable for an increasing number of human activities; for this reason, it is difficult to imagine a modern society without plastics because of its uses in several areas such as construction, packaging, medicine, electronics, etc. [1]. Unfortunately, the intensive production and use, as well as the inadequate disposal of plastic waste, have led to serious environmental problems. In 2017 it was estimated that the annual production of potentially harmful plastics waste exceeds 150 million tons worldwide, which directly or indirectly affects the health of most living beings as well as the natural balance of ecosystems [2]. Due to the environmental concerns caused by the accumulation of plastics wastes, academic, industrial and government sectors are developing strategies to increase the sustainability of the plastic industry including the reduction of plastics in single-use applications, recycling and the production of more sustainable materials such as composites based on biobased fillers (flax, hemp, pine, coir or agave fibers) [3,4,5].

In the last decades, the use of natural fibers as reinforcement of thermoplastics has drawn a great deal of attention due to their interesting specific mechanical properties, low environmental impact and cost, wide availability, biodegradability and lightweight (low density) compared to synthetic fibers (glass or carbon) and other fillers (calcium carbonate) which are still widely used in the industry [6]. However, the main limitation of natural fibers is their low compatibility with most hydrophobic polymers, leading to poor filler-matrix adhesion. They are also susceptible to moisture absorption, which affects the dimensional stability and overall performance of the composites [7]. Nevertheless, under optimized processing parameters, they can be processed via injection [8], compression [9] and rotational molding, with properties and design suitable for any particular application [10].

Rotational molding is a processing technology suitable to produce hollow parts from small ping-pong balls to large (up to ten thousand liters) tanks [11,12]. Due to its unique characteristics such as low shear and low waste production, lower capital investment costs, lower restrictions on mold design and easier production of multi-layer materials, especially when different composition and thicknesses are required, rotational molding has several advantages over other processing methods. In rotational molding, multilayered materials can be easily and individually reinforced with different fillers (composites) and/or lightened using different foaming agents, as per the application requirements, client specifications and/or design requirements, in terms of performance and aesthetics [13].

Although limited information is available on this process, some studies on rotational molding with natural fibers have been found. Torres and Diaz conducted experiments with natural fibers (jute, sisal, cabuya and coir) with high density polyethylene (HDPE) as the matrix [14]. Ward and Rodrigue used pine fibers with linear medium density polyethylene (LMDPE) [15]. Lopez-Bañuelos et al. [16] conducted a study on rotational molding using dry-blending with agave fibers (0–15 wt %) and LMDPE as the matrix. Although these single-layer systems are commonly found in the literature, it should be noted that the advantages of rotational molding allow a growing interest in producing multilayer materials for several specific applications, as addressed in this study. In this sense, Torres and Aragon produced bilayer composites via rotational molding using HDPE and sisal with a maximum concentration of 7.5 wt % [17]. The fiber was used in the external layer (first layer), while HDPE alone was in the inner layer (second layer). They observed that the composite reinforced with 5 wt % of sisal fiber produced the best properties as the tensile modulus increased by 20% while the tensile strength remained almost constant (14 MPa), compared to neat polyethylene. However, the effect of the two-layer configuration was more significant on the impact properties as the impact energy decreased from 6.5 to 2.9 J due to fiber addition. This effect was related to the addition of fibers inducing a fragile fracture behavior caused by a poor fiber-matrix adhesion (local defects) and the rigidity of the fibers themselves limiting the polymer chain mobility.

Ortega et. al. [18] studied the behavior of two and three-layer structures produced by rotational molding using a metallocene polyethylene (mPE) with banana and abaca fibers (5 wt %). The impact properties (Gardner) were significantly reduced due to the fiber content: an 87% decrease of impact strength was observed for the two-layer structures and a 68% reduction for three-layer materials, compared to the neat matrix. Nevertheless, significant increases in tensile modulus (188%) and flexural modulus (200%) were observed.

For rotomolded foamed materials [19], several investigations have been published due to the interesting change in mechanical properties and density. Moscoso et al. [20] produced a foamed single-layer material based on LMDPE using azodicarbonamide (ACA) as a chemical blowing agent (0 to 1 wt %). They observed significant differences between the foamed and unfoamed materials, especially for the total processing cycle time and the final density of the rotomolded parts. For example, the density decreased by 68% with 1 wt % ACA. Liu and Yang studied how the processing conditions influenced the properties of linear low-density polyethylene (LLDPE) foams for two-layers materials [21]. In the external layer, polyethylene was used alone while the inner layer was foamed. The results showed some asymmetry according to the tested surface; the impact strength was higher when the materials were impacted on the external face (unfoamed layer).

Casavola et al. [22] studied the low impact behavior of three-layer materials produced by rotational molding made from two polyethylene layers enclosing a foam layer. The results showed that the average force to penetrate the external face was about 28% higher compared to the internal face, which is related to the compression behavior of the foam absorbing a part of the impact energy and also reducing the plastic deformation of the skin (force needed to penetrate in the layer), which is convenient for some applications. One of the strengths of this work is the use of agave fibers (an agro-waste of tequila production in Mexico) for the processing and formulation of bilayer materials using a composite in the external layer and a polymer foam in the internal surface, in order to increase the sustainability by reducing the cost/weight ratio of the parts and the dependence of oil-based polymers. As reported above, these systems are commonly composed only by polyethylene and foamed polyethylene.

As a preliminary work, three-layer materials were prepared with agave fibers as reinforcement, azodicarbonamide as a foaming agent and LMDPE as the matrix [23]. In the external layer, agave fibers were used, while the middle layer was foamed using ACA and the inner layer was neat LMDPE. Via morphological analyses, gas diffusion from the blowing agent decomposition in the middle layer to the external layer was detected, but this migration was not detected towards the inner layer. This effect was related to the addition of agave fibers whose porous nature could have facilitated the diffusion of gases to the external layer. In general, the density decreased with the ACA content. These behaviors led to changes in the impact properties as the impact strength decreased with both agave fiber (AF) and ACA contents due to the poor filler-matrix adhesion. A linear relation between impact strength and relative density was observed. In addition, it was reported that the internal layer (foamed polyethylene) collapsed due to the intensive heating of the three layers, which also increased the processing time; for that reason, in this project a double layer system is proposed.

In this work, the main objective was to prepare and characterize bilayer rotomolded materials based on foams and composites using linear medium density polyethylene as the matrix. Agave fibers (0 to 15 wt %) were used in the external layer (first layer) to produce a composite (to improve mechanical properties), while azodicarbonamide (0 to 0.75 wt %) was added in the inner layer (second layer) to produce a foam (density and weight reduction). The novelty of this work was to determine the effect of agave fiber and azodicarbonamide content on the internal air temperature profile and perform a complete characterization of the samples in terms of thermal history, morphological and mechanical properties. In addition, a wide range of AF and ACA content was added, using a simple dry-blending technique, which in addition to providing an alternative for reuse of agave fibers from the production of tequila, allows the production of lighter materials at lower cost with the possibility of controlling the properties, performance and aesthetics (appearance) of the rotomolded parts.

## 2. Materials and Methods

### 2.1. Materials

The matrix used was linear medium density polyethylene (LMDPE) RO 93650 from Polímeros Nacionales (Tlaquepaque, Mexico) with a density of 0.936 g/cm^3^ and a melt flow index of 5 g/10 min (2.16 kg/190 °C). Agave fibers (AF) (*Agave tequilana* Weber var. Azul) were obtained from the residues of a local tequila industry in Jalisco (Tequila, Mexico). To generate the foamed layer, azodicarbonamide (ACA) as a chemical blowing agent (CBA), was provided from Grupo ALBE (Guadalajara, Mexico) with a decomposition temperature between 170 and 220 °C.

### 2.2. Fiber Preparation

The agave fibers were first placed in water for 24 h to remove impurities. Then, the fibers were washed with water in a double-disc (30 cm diameter) refiner Sprout-Waldron (D2A509N) with one rotating disc (1770 rpm) while the other was stationary in order to remove pith. The fibers were subsequently centrifuged to remove as much water as possible before being sundried for a minimum 2 days (until a moisture content below 10% was reached). Finally, the fibers were grounded and sieved to keep only the fibers between 100 mesh and 140 mesh to give and average size between 106 and 150 microns. This particle size produced the best dispersion, distribution and mechanical performance, according to previous studies [16,23]. Before rotational molding, the fibers were dried again in an oven for 24 h at 60 °C to reach moisture contents less than 1%. The average aspect ratio (length/diameter = L/D) of the fiber was 2.43.

### 2.3. Rotational Molding

The samples were produced in a laboratory-scale rotational molding machine with a cylindrical stainless-steel mold of 3.6 mm in thickness, 19 cm in diameter and 25.5 cm in length to give a total volume of around 5 L. The materials for each layer were initially dry-blended before their introduction into the mold. This was performed using a Torrey LP-12 mixer for 5 min at 1750 RPM. For the composite layer, different agave fiber contents (0, 5, 10 or 15 wt %) were used. The blends were again dried overnight in an oven at 60 °C before processing. Figure 1 shows the production sequence of bilayer rotomolded materials. To form the first layer, the polymer/fiber mixture was placed in the mold which was introduced into the oven previously heated at 260 ± 5 °C with a speed ratio of 1:4 for 15 min. Subsequently, the mold was removed from the oven to load the material for the second layer composed of the LMDPE matrix with different ACA contents (0, 0.15, 0.25, 0.50 or 0.75 wt %) using the same processing conditions as for the first layer (dry-blending, temperature and speed). For each layer, 270 g of material was used. Finally, the mold was taken out of the oven to cool down by forced air convection. The internal air temperature (IAT) was monitored by a thermocouple placed inside the mold through the vent conduct during processing. Cooling was applied until the IAT dropped to 40 °C before demolding the parts. Finally, the samples were cut in order to obtain the test specimens for each characterization as described next.

### 2.4. Characterization

#### 2.4.1. Thermogravimetry (TGA)

Thermal decomposition was followed in terms of weight loss to evaluate the thermal stability of the raw materials. The tests were performed on a model Q5000IR from TA Instruments (New Castle, DE, USA). Samples between 10 and 20 mg were analyzed with a heating rate of 10 °C/min between 50 and 500 °C. To evaluate the thermal and oxidative resistance under a nitrogen atmosphere.

#### 2.4.2. Morphology

The rotomolded parts were cryogenically fractured with liquid nitrogen and their exposed surfaces were covered with a layer of Au/C for one minute. Then, micrographs were taken on a scanning electron microscope (SEM) TESCAN model MIRA3 (Warrendale, PA, USA) at different magnifications. To obtain a complete morphological analysis, images were also taken on an optical microscope Olympus MIC D (Mexico City, Mexico). For thickness, image analysis via the Image-Pro Plus 4.5 (Media Cybernetics) software was performed. A line was drawn in a minimum of 10 images to report the averages and standard deviations.

#### 2.4.3. Density

Density was calculated from the weight and the linear dimensions (thickness, width and length) of rectangular sections of the samples. The results reported are the average of a minimum of 10 samples with their respective standard deviation.

#### 2.4.4. Mechanical Properties

All the specimens for mechanical tests were cut from the rotomolded parts in the longitudinal direction of the cylinder (as shown in Figure 1), in order to avoid the curvature of the geometry of the piece. Finally, the specimens were tested at room temperature. All the samples were analyzed after one week to allow for stress relaxation before testing.

Charpy impact strength was determined by an Instron Ceast model 9050 (Norwood, MA, USA) impact tester. The specimens were prepared according to ASTM D6110 with dimensions of 80 × 12.7 mm^2^ (the thickness changed with the AF and ACA concentration). All the samples were notched in the center of the longitudinal side by a manual notching machine Instron Ceast 6897 (Norwood, MA, USA). The values reported represent the average and standard deviation of ten repetitions.

Gardner impact tests were carried out using a CSI falling weight impact tester model 285 (Whippany, NJ, USA) according to ASTM D5420. Sample dimensions were related to the concentration of fiber and foaming agent according to the standard: the width was less than 25 mm and the thicknesses variation for each composition was less than 5%. A total of 20 specimens for each sample was analyzed and the values reported are the average and standard deviation of each condition.

Tensile properties were measured on a universal testing machine Instron model 4411 (Norwood, MA, USA) with a 1000 N load cell. Dog-bone type V samples were cut from the molded parts according to ASTM D638. The crosshead speed was set at 5 mm/min. The stress-strain curves were analyzed to get the modulus, strength and elongation at break based on the average of five samples with their standard deviation.

Flexural tests were performed on an Instron universal testing machine model 3345 (Norwood, MA, USA) with a 1000 N load cell according to ASTM D790 using a crosshead speed of 0.1 mm/min. Sample dimensions were 80 × 12.7 mm^2^ (the thickness changed with the AF and ACA concentration). At least five samples were used to report the average and standard deviation for modulus and strength.

For Gardner impact strength and flexural properties, the tests were carried out on both the internal (foam) and external (composite) surfaces of the rotomolded parts for a more complete analysis with respect to the asymmetric structure produced.

## 3. Results and Discussion

### 3.1. Thermogravimetry Analyzes

Figure 2 presents the thermal decomposition of the raw materials. The degradation of LMDPE started at around 420 °C and finished around 500 °C (Figure 2a) indicating good thermal stability for the range of conditions used in rotational molding (oven temperature of 260 °C). For ACA, the decomposition began at around 190 °C and was completed by 240 °C [24], where the peak decomposition occurred at 224 °C (Figure 2b). For the agave fibers, an initial mass loss of about 5% between 35 and 100 °C was related to fiber dehydration [25]; the main mass loss was in the range of 250–390 °C where the thermal degradation of hemicellulose (304 °C) and lignin (376 °C), the main components of the fibers, occurred [26].

### 3.2. Processing Cycle Time

Figure 3 presents the internal air temperature (IAT) during rotational molding as a function of time for an oven temperature of 260 ± 5 °C. From these curves, Table 1 reports on the values of peak internal air temperature (PIAT) and on the total cycle time for the different AF content and 0.75 wt % of ACA. The PIAT values of layer 1 did not show a significant change with the relatively low amount of fiber used in the present work. Conversely, Mounika et al. [27] reported variations of the thermal conductivity as a function of the fiber content, while Banerjee et al. [28] reported that increasing the fiber content decreased the heat transfer rates and modified the PIAT values. When the material was added for the second layer, there was a significant increase in its PIAT (Table 1) due of the longer time in the mold (20 min for layer 1 versus 32 min for layer 2). When the ACA was present the PIAT increased by about 2 °C because of the exothermic reaction of the ACA decomposition. Nevertheless, PIAT variations can be directly related to the materials and processing conditions and is very sensitive to differentiate between the samples as reported in other studies [21,29,30,31].

The thermogravimetry analysis of the agave fiber (Figure 2b), explains why there was no thermal degradation of the natural reinforcement since the maximum temperature inside the mold (PIAT) was around 174 °C for all the compositions. The analysis of processing cycle time and internal air temperatures are key parameters due to the lack information available in the literature and their industrial significance. Rotational molding has longer cycle times, compared to other processing technologies, the raw materials are susceptible to suffer thermal degradation, particularly natural fibers. From these data, it was possible to confirm that the adequate processing conditions (e.g., oven temperature and heating times) were used, which guarantees the correct sintering process of the part, avoiding the thermal degradation of both fiber and matrix.

### 3.3. Morphology

To determine the total thickness of the bilayer samples, a series of cross-sectional measurements were made using optical microscopy. Figure 4 clearly shows the formation of both layers. Figure 4a,c show that by increasing the ACA content larger total thickness was obtained. Figure 4b,d show that the presence of fibers in the first layer favors the formation of bubbles in this layer, thus increasing the total thickness.

Figure 5 shows that the total sample thickness increases with increasing foaming agent content in layer 2. An increase of 83% in the total thickness was obtained with the maximum ACA content used (0.75%) compared to the same part without ACA (0%). For three-layer materials obtained by rotational molding using a foaming agent where a thickness increase of 56% was reported [23]. When processing by rotational molding single-layer materials, a thickness increase from 3 to 15 mm was observed when 1 wt % ACA was used. This increase is related to the high amount of gas released by ACA as 1 g of ACA produces 228 mL of gas [20]. As seen in Figure 5, when ACA was added, the amount of fiber in the first layer affects the total thickness. The addition of 0.75% ACA resulted in 103%, 108% and 127% thickness increase when using 5%, 10% and 15% of AF, respectively. In a previous work [23], it was shown that at low proportions (<15 wt %), fiber addition generally does not affect the total thickness of rotomolded parts.

Typical scanning electron microscope (SEM) images of the bilayer samples are presented in Figure 6, where the formation of two layers is clearly seen in Figure 4. In the presence of AF, the gas produced by the ACA diffuses from the second layer to the first layer because of the long cycle time and high temperature. This gas diffusion generates a transition layer, and this is why a sharp interface is not observed in the optical images (Figure 4a,c) and in the SEM images (Figure 5a,c). This transition can be mechanically interesting as creating a density/composition gradient that can remove stress concentration point and properties discontinuities as reported for density gradient materials (DGM) [19].

The morphological analysis also shows a large number of voids at the fiber-matrix interface (Figure 7) due to the hydrophilic nature of AF compared with the hydrophobic nature of LMDPE. These voids, as well as the natural porosity of AF promote the migration of gas molecules to regions having higher temperature and void content [32]. This structure also helps bubble nucleation and growth in the outer layer (close to the mold wall) from the heterogeneous nucleating effect of the fibers acting as nucleating agents [8,33,34]. The presence of bubbles in the first layer when using ACA also explains the thickness increase as reported in Figure 5. Since AF particles can act as nucleating agents, the cell structure becomes a complex function of both parameters (AF and ACA content). Nevertheless, some of the smaller voids seen in Figure 7 can be related to fiber pull-out [5].

Table 2 presents the density of the foamed and unfoamed samples. As reported in previous works [9,35], the density decreased with increasing ACA content due to the higher amount of gas generated. The density of unfoamed samples (without ACA) increased with fiber content, because the AF cell-wall density (1.59 g/cm^3^) was higher than that of LMDPE (0.93 g/cm^3^) [6,36]. However, this behavior was not observed when ACA was used because of the interfacial voids created at the fiber-matrix interface as well as the fibers acting as nucleating agents (more efficient foaming process). For LMDPE (0.93 g/cm^3^), the density decreased by 41% when the maximum ACA concentration (0.75%) was used; and the density decreased by 44%, 47% and 55% with the addition of 5, 10 and 15 wt % of AF, respectively.

### 3.4. Impact

Figure 8 presents the Charpy Impact strength as a function of ACA and AF concentrations. As expected, the foamed parts showed lower impact strength than the unfoamed parts [21]. Due to the foam effect, the LMDPE impact strength decreased by 31% (from 94 to 65 J/m). Due to the addition of AF, the impact strength decreased by 36% (from 90 to 58 J/m) at 5 wt % AF. Similarly, this decrease was 47% (from 86 to 46 J/m) and 52% (from 84 to 40 J/m) with 10 and 15 wt % AF, respectively. For the samples with the maximum concentrations of AF and ACA, the impact strength decreased by 57% compared to the neat LMDPE, probably due to some matrix discontinuities caused by the bubbles (foam), the poor adhesion between the fiber and the matrix (Figure 7), as well as the presence of rigid particles (AF) inducing a brittle fracture behavior (easy crack initiation and propagation) which may explain the low impact resistance observed. The bubbles generated by the ACA were expected to collapse under deformation when subjected to external impact forces leading to reduced impact energy absorption. Similar results were reported in the literature [17,18].

The results of Gardner’s impact strength test on the outer surface and the inner surface as a function of ACA and AF content are shown in Figure 9. The Gardner impact resistance of the neat LMDPE showed no significant differences by applying the impact on either side. A similar result was reported by Tovar et al. [37] in foamed materials made by injection molding. However, for materials when different fiber concentrations were used, when the impact was on the external surface (Figure 9a) the impact strength was slightly lower than when the impact was on the internal surface (Figure 9b). The literature reports that higher impact resistance was obtained when a foamed sample is impacted on the side having the thinner skin since the energy is distributed in a greater volume of material [20]. Nevertheless, the impact strength tended to decrease when adding both AF and ACA.

Gardner’s impact strength for the unfoamed LMDPE was 7,350 J/m, and a reduction of 42% was observed when 0.75 wt % ACA was used. When adding 5% AF, the impact resistance decreased by 16% when the impact was made on the external face (composite surface) and by 13% on the internal face (foam surface) compared to the unfoamed LMDPE. Similar results were reported by Ortega et al. for two-layer materials produced with metallocene polyethylene (mPE) and 5 wt % of abaca fiber [18]. When impacted on the outer surface, a reduction of 28% and 34% was observed by adding 10 and 15% in weight of FA, respectively, values that decreased to 22% and 26% when impacted on the inner surface, respectively. When the maximum amount of ACA (0.75%) and AF (15%) was used, the impact resistance decreased by 56% (external surface) and 60% (internal surface) compared to the neat matrix. It can be concluded that the matrix became more brittle, and the ability of the material to absorb impact energy decreased as AF content increased. Higher AF content also led to more fiber agglomerations and interfacial voids acting as stress concentrator points helping crack propagation within the structure [17]. As mentioned above, this is related to poor interfacial adhesion between the fibers and the matrix [16]. To improve the interfacial interaction/adhesion, coupling agents and fiber surface treatments can be used [38]. When comparing the impact resistance results (Figure 8 and Figure 9) with the composite density (Table 2), it can be noticed that the lower the density, the lower the impact resistance.

### 3.5. Tension and Flexion

Figure 10 shows the tensile modulus for the bilayer materials. As expected, an increase in tensile modulus was observed for composites containing AF but without ACA compared to the neat LMDPE (155 MPa). The modulus increase is similar to those reported in the literature for fiber-reinforced rotomolded polymers: 10% for single layer materials [16,36], 18% for bilayer materials and 37% for three-layer materials [18]. These expected improvements are related to the presence of a rigid phase (AF) in the matrix [10]. This increase was still present even with 0.15% and 0.25 wt % ACA was used: 21% and 6% of increase, respectively with 15 wt % AF. However, this improvement was not observed at higher ACA contents: the modulus dropped by 28% and 34% at 0.50% and 0.75 wt % ACA content respectively, due to the higher porosity of the samples. As reported in the literature, the modulus decreased with increasing foaming agent content, as the density decreased; i.e., less material per unit volume was available to distribute the applied stresses [20].

Tensile modulus reduction up to 45% were obtained compared to the neat LMDPE, while values of 54%, 64% and 70% for the 5, 10 and 15 wt % AF, respectively. It was observed that the combination of AF and ACA plays a very important role on the results obtained. As seen in Figure 7, the fiber-matrix interaction was controlling the voids generated which had a direct effect on the density (Table 2); lower density was obtained when both AF and ACA content increased. Finally, voids were discontinuities creating some defects in the materials leading to lower mechanical properties [39].

Tensile strength as a function of ACA and AF is presented in Figure 11a. In all cases, the tensile strength did not present improvements with AF and ACA addition, mainly because of density reduction and presence of voids (fiber-matrix incompatibility). For unfoamed materials, tensile strength was reduced by 13% at the maximum AF content (15%) compared to the neat LMDPE. When ACA was used, decreases of 14–52% on tensile strength were observed for the range of conditions studied, a similar range of values was obtained when AF was added: 17–57% (5%), 20–56% (10%) and 25–56% (15%). This decrease can be explained by the weak interfacial adhesion associated with the hydrophilic characteristics of the fibers compared to the hydrophobic nature of the matrix. For the foamed materials, lower tensile strengths were observed with increasing ACA content; i.e., when generating a greater amount of porosity (gas bubbles) inside the material [7]. Similar results have been reported in the literature [16,36].

Figure 11b shows the elongation at break of the bilayer materials. As mentioned above, the adhesion as well as the distribution of AF in the matrix are key factors to control the final properties of these materials. As expected, a significant reduction was observed when adding AF or ACA to the LMDPE. For example, the reduction in elongation at break reached 83% at 15 wt % AF and 0.75 wt % ACA compared to the neat LMDPE. This decrease is related to the low elasticity of AF (rigid filler) limiting the motion of long macromolecular chains [16,38]. A similar elongation reduction occurred when ACA was used because of the presence of the generated voids. As for most polymer composites, fiber addition provided rigidity/strength but limited ductility/toughness [40]. A similar elongation reduction occurred when ACA was used because of the presence of the generated voids.

A comparison between the flexural (F) and tensile (T) moduli as a function of AF content is shown in Figure 12. As expected, the flexural modulus was higher than the tension moduli (66% for LMDPE and 70% for AF). Similar results were obtained for a single layer [36] and three-layer materials prepared by rotational molding [18]. The maximum value obtained for flexural modulus was 638 MPa (with 15% AF), which represents a 29% increase compared to the neat LMDPE. Unfortunately, these increases were not obtained when ACA was used as the modulus decreased between 3% and 27% for the neat LMDPE and between 8% and 56% when 15 wt % AF was used. This decrease follows the same trend as for the results in tension and as reported by Archer et al. [41], because there was less material mass per unit of volume available to support the loads (density decrease) [32].

## 4. Conclusions

In this work, bilayer materials were produced via rotational molding using linear medium density polyethylene (LMDPE) as the matrix. The results showed that negligible variations were observed for the peak internal air temperature (PIAT) of the first layer, but the PIAT values were increased significantly for the second layer. This was attributed to the longer time on the mold. 

The thickness of each layer was found to increase with both AF (more void and defects) and ACA (because foam) content, leading to a 55% density reduction (0.93 g/cm^3^) compared with that of the neat matrix (0.44 g/cm^3^). The poor interfacial adhesion between fiber and matrix is due to their incompatibility and the voids that reduce the mechanical properties. Impact strength of LMDPE decreased by 31% (from 94 to 65 J/m), and 52% (from 84 to 40 J/m) with 15 wt % AF. Bubbles collapsed under deformation when subjected to external impact forces leading to reduced impact energy absorption.

Tensile test showed an increase of 23% on Young’s modulus and a reduction on tensile strength with the incorporation of AF up to 13% compared to the neat LMDPE. With the incorporation of ACA, Young’s modulus and tensile strength was reduced. When ACA and AF were added simultaneously, the modulus of the LMDPE was increased. However, with an ACA content higher than 0.25%, even with AF the tensile modulus decreased. Similar results were obtained for the flexural modulus. A large reduction on elongation at break was observed when adding AF or ACA to the LMDPE. Based on the results of this work, it is concluded that it is possible to manufacture tailor-made multilayer parts containing agave fiber and azodicarbonamide, by rotational moulding. The properties of the bi-layers materials are a balance between the reinforcement effect of the fibers and the weight reduction of the foams. Nevertheless, more work needs to be done, especially to improve the fiber-matrix interfacial adhesion and to determine its effect on the processing and performance of the rotomolded parts, e.g., limiting the gas migration. The production of bi-layer materials based on agave fibers and foams allowed to produce lighter materials with potentially lower costs and to control the specific properties, performance and aesthetics (visual aspect) of the rotomolded parts.

## Figures and Tables

**Figure 1 polymers-12-00503-f001:**
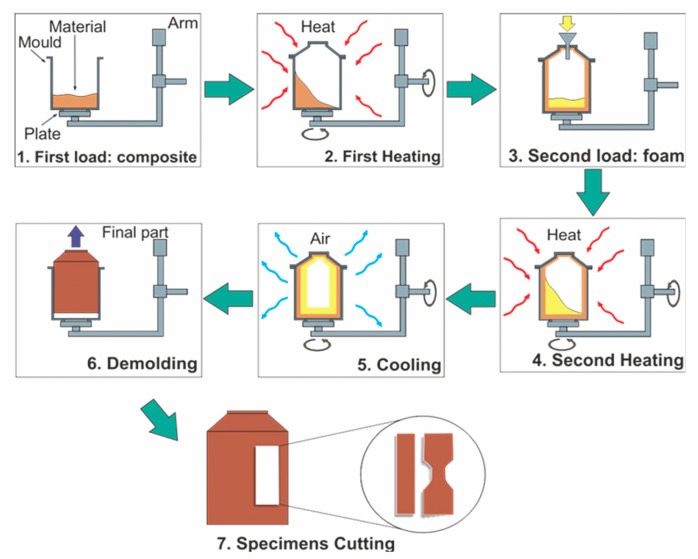
Production sequence of bilayer rotomolded materials.

**Figure 2 polymers-12-00503-f002:**
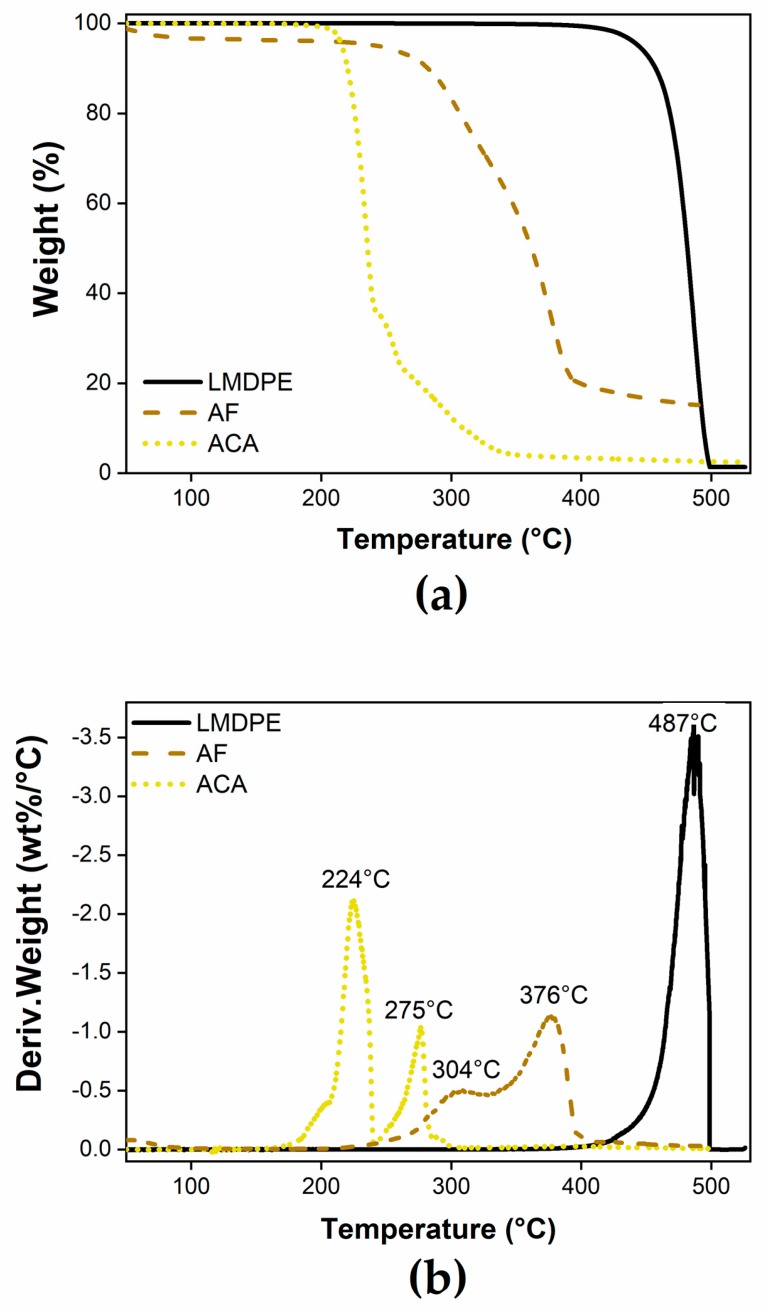
TGA (**a**) and TGA derivative (**b**) curves of the neat materials.

**Figure 3 polymers-12-00503-f003:**
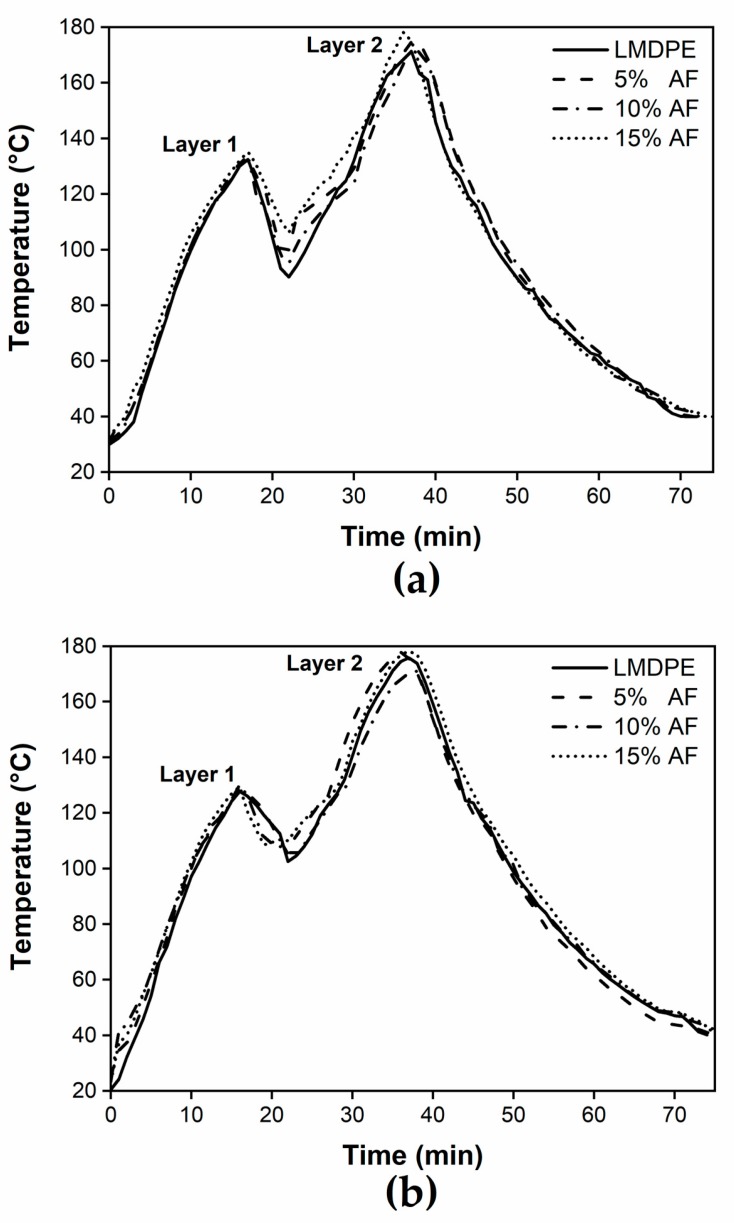
Internal air temperature (IAT) profiles during the rotational molding process for samples with different azodicarbonamide (ACA) content: (**a**) 0 wt % and; (**b**) 0.75 wt %.

**Figure 4 polymers-12-00503-f004:**
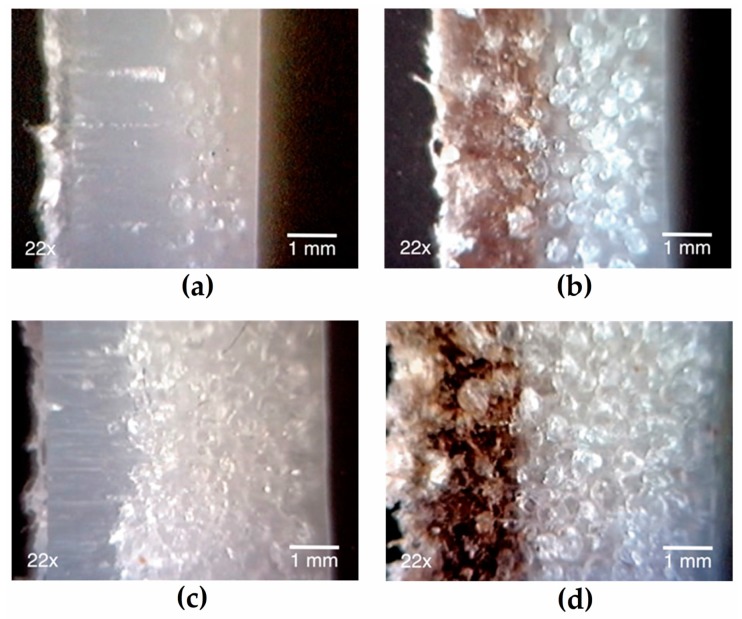
Typical optical images of the different bilayer samples cross-sections, linear medium density polyethyle (LMDPE) with (layer 1, wt % AF/layer 2, wt %ACA): (**a**) 0/0.25; (**b**) 15/0.25; (**c**) 0/0.75; and (**d**) 15/0.75.

**Figure 5 polymers-12-00503-f005:**
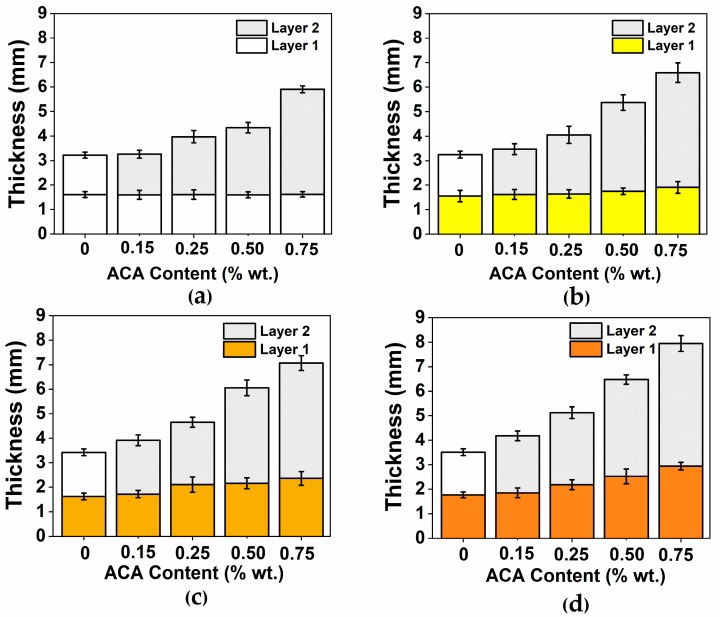
Layer thickness as a function of ACA content for samples with different fiber concentrations: (**a)** 0; (**b**) 5; (**c**) 10; and (**d**) 15 ( wt %).

**Figure 6 polymers-12-00503-f006:**
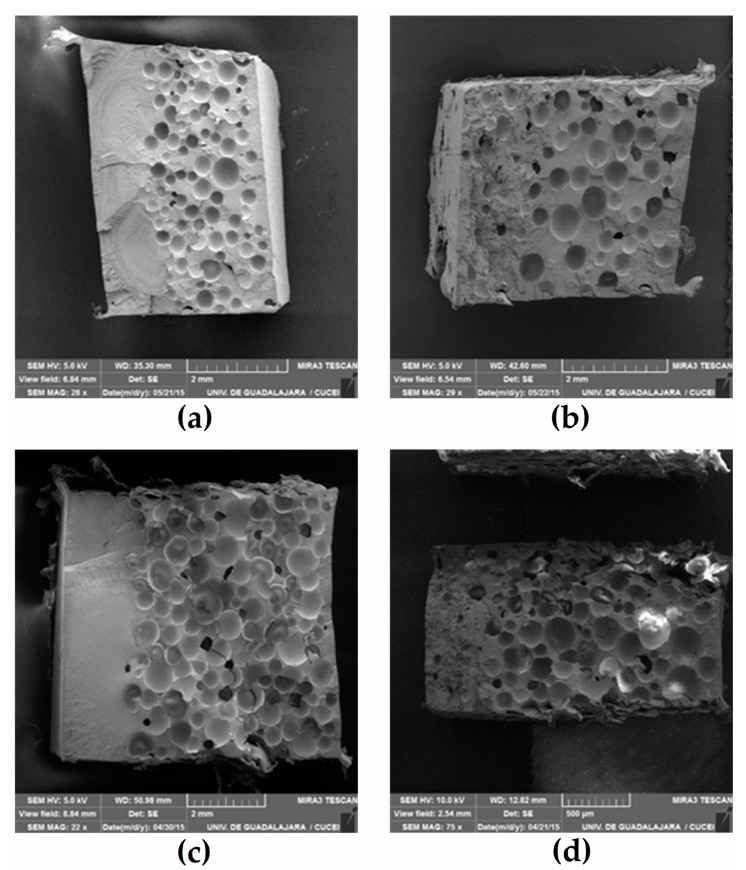
Typical SEM micrographs of the different types of bilayer samples produced, LMDPE (wt % agave fiber (AF)/ wt % ACA: (**a**) 0/0.25; (**b**) 15/0.25; (**c**) 0/0.75; and (**d**) 15/0.75.

**Figure 7 polymers-12-00503-f007:**
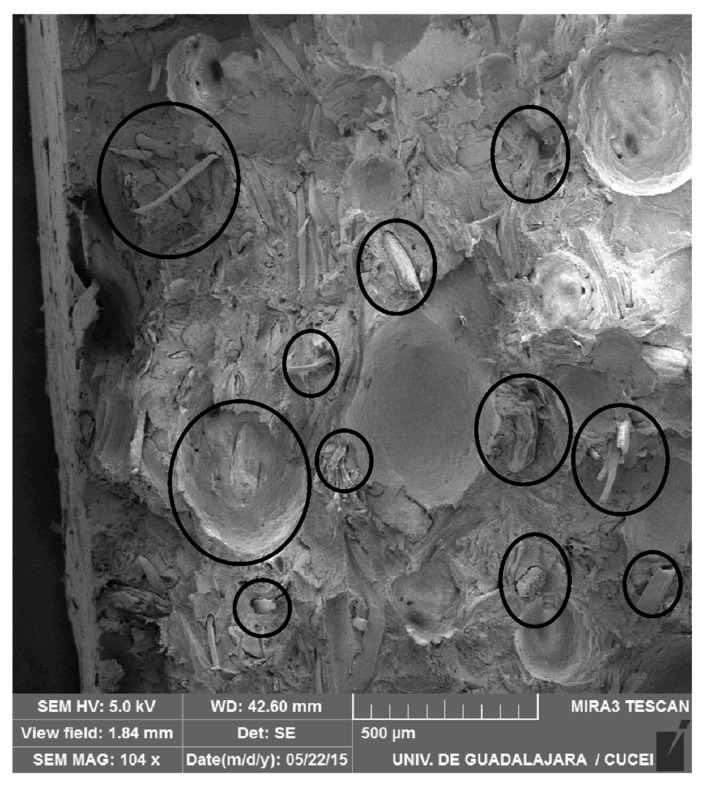
Typical morphology of the rotomolded samples (external layer of the bilayer sample) showing the fiber distribution and the voids at the fiber-matrix interfase.

**Figure 8 polymers-12-00503-f008:**
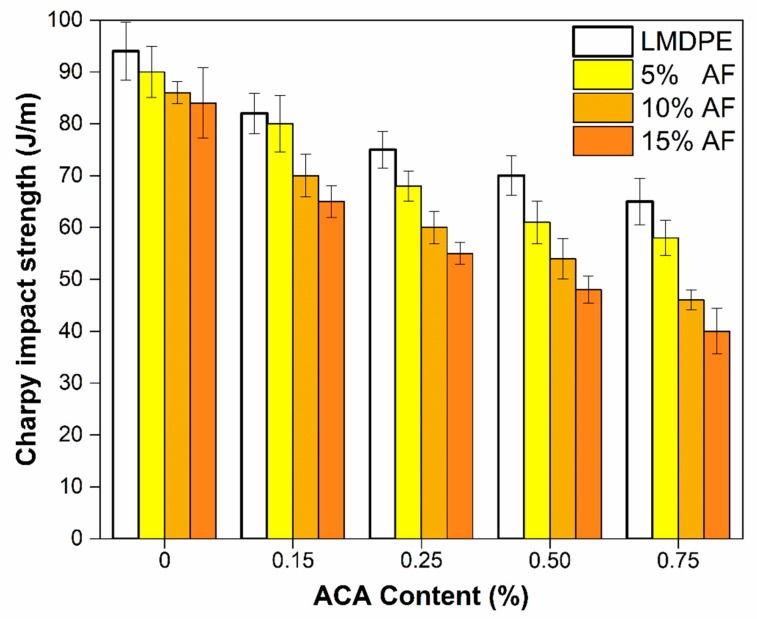
Charpy impact strength of the bilayer samples.

**Figure 9 polymers-12-00503-f009:**
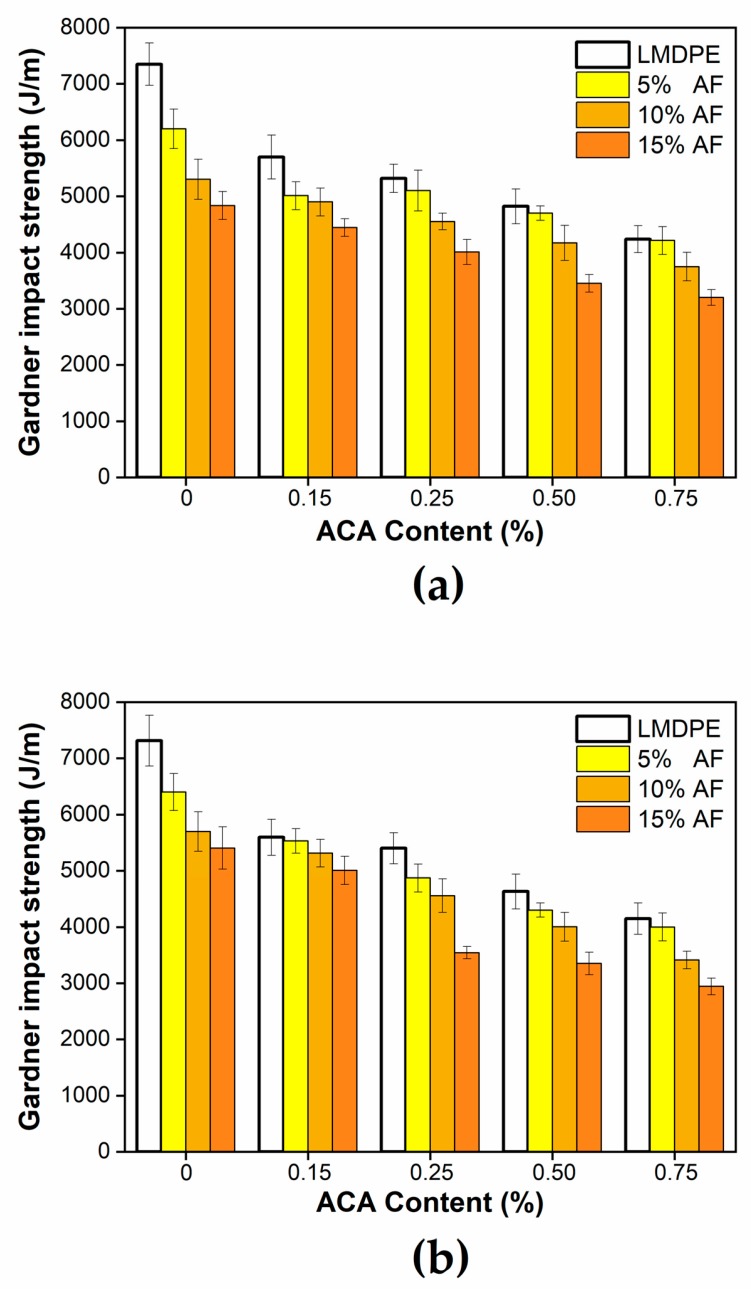
Gardner impact strength of the bilayer samples for an impact made on the: (**a**) external surface; and (**b**) internal surface.

**Figure 10 polymers-12-00503-f010:**
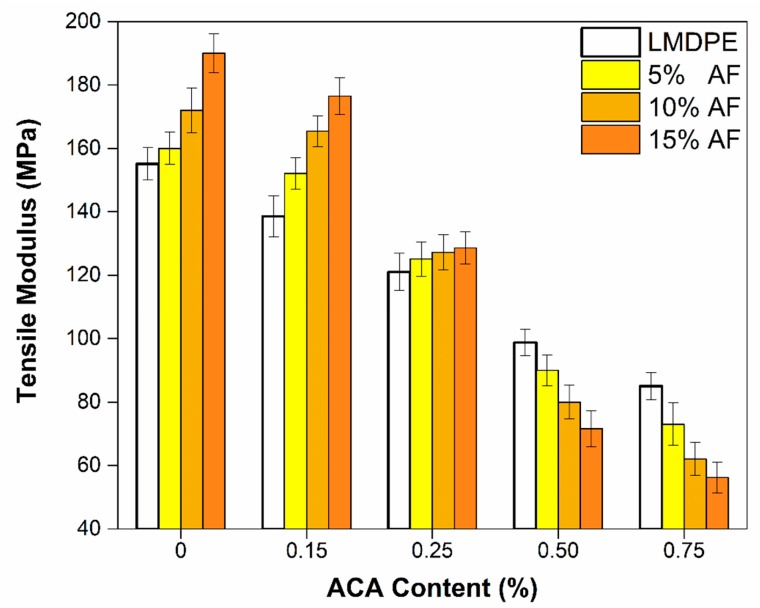
Tensile modulus of the bilayer materials as a function of ACA and AF contents.

**Figure 11 polymers-12-00503-f011:**
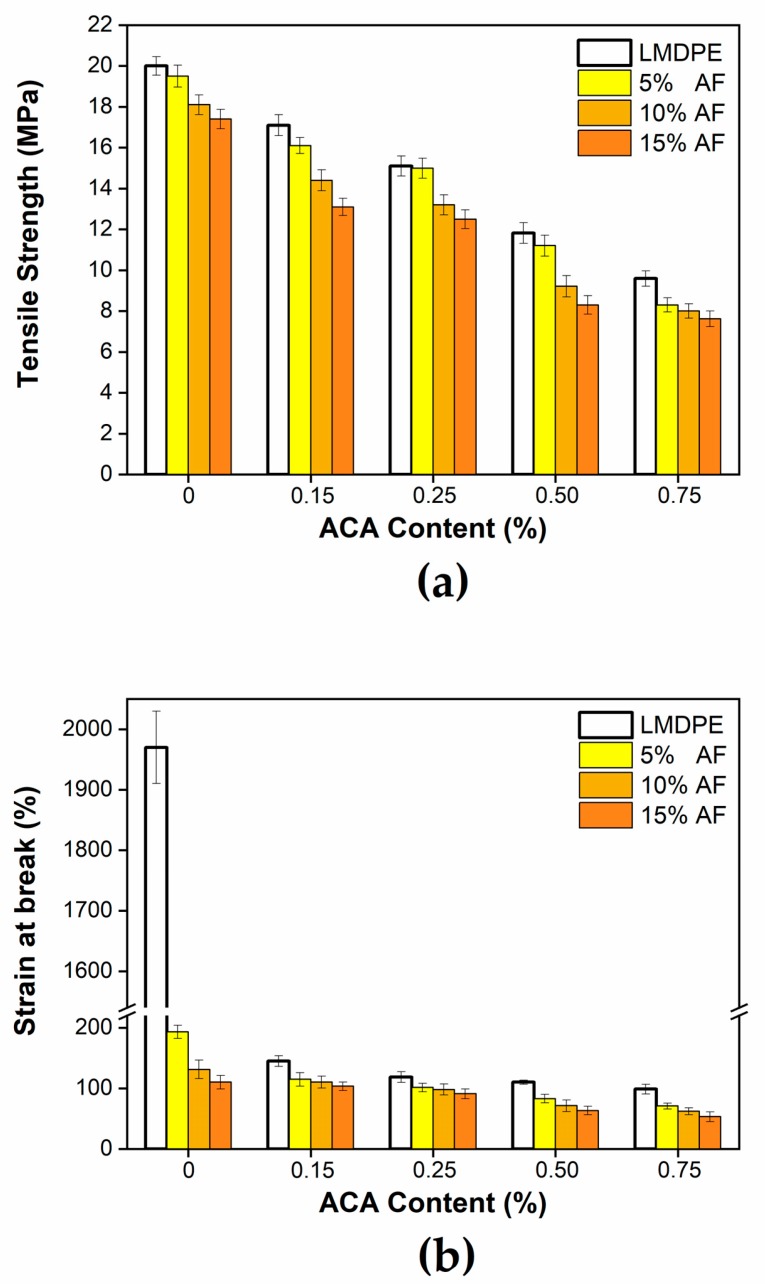
Tensile strength (**a**) and elongation at break (**b**) of the bilayer materials as a function of ACA and AF contents.

**Figure 12 polymers-12-00503-f012:**
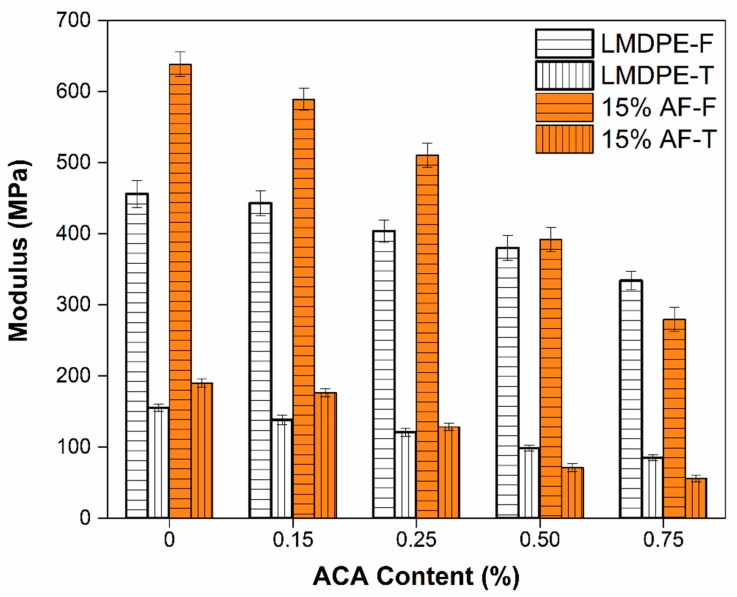
Flexural and tensile moduli of LMDPE and 15 wt % AF bilayer materials as a function of ACA content.

**Table 1 polymers-12-00503-t001:** PIAT values and total cycle time for the rotomolded samples with 0% (without foam) and 0.75% (foamed) ACA.

Fiber Content (wt %)	PIAT (°C) Layer 1	PIAT (°C) Layer 2		Total Cycle (min)	
		Without Foam	Foamed	Without Foam	Foamed
0	128	171	174	71	74
5	129	173	174	72	75
10	130	171	172	73	76
15	129	176	178	74	77

**Table 2 polymers-12-00503-t002:** Density (g/cm^3^) of the bilayer materials.

ACA Content (wt %)	Fiber Content (wt %)			
	0	5	10	15
0	0.93 ± 0.08	0.94 ± 0.05	0.96 ± 0.07	0.98 ± 0.06
0.15	0.80 ± 0.07	0.82 ± 0.06	0.86 ± 0.08	0.90 ± 0.07
0.25	0.71 ± 0.09	0.68 ± 0.06	0.65 ± 0.05	0.64 ± 0.03
0.50	0.64 ± 0.06	0.61 ± 0.06	0.55 ± 0.06	0.51 ± 0.06
0.75	0.55 ± 0.05	0.53 ± 0.02	0.51 ± 0.09	0.44 ± 0.08

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
