# Peer review of "Morphological and Mechanical Properties of Bilayers Wood-Plastic Composites and Foams Obtained by Rotational Molding"

_polymers, 2020, doi:10.3390/polym12030503_

Round 1

Reviewer 1 Report

This manuscript reported their findings on the morphological and mechanical properties of bilayers formed LMDPE composite reinforced with agave fiber. The manuscript is readable, and the research results can provide a reference for industrial production. In my opinion, it can be accepted after following revisions.

  1. Although the authors reviewed some recent researches in the introduction part, which is good, the objective of their investigation is not clear enough. The authors should give a summary in each review paragraph and explain how these studies are relevant/differ to current study.
  2. Line 91, if the “rotomolded” is same to “rotational molding”, please uniform the appellation.
  3. Please give some explanations on the employment of agave fibers and why use the fiber size of between 100 mesh and 140 mesh.

    4. I have some confuse about the “Rotational Molding” processes, could you please show some detail about the mold, pictures or schema. What is the mold temperature and pressure during the molding process? How to composite the two layers?

  1. “3.2. Processing Cycle Time”, this part of the analysis is a bit superficial and redundant, and is more suitable for presentation in the materials and methods section.
  2. line 385, “FA”? Is it “AF”?

Author Response

We would like to acknowledge Reviewer 1 for their constructive comments that led us to improve our manuscript. Following the suggestions, the manuscript has been carefully edited and revised. All changes were typed in blue color.

Comment 1: This manuscript reported their findings on the morphological and mechanical properties of bilayers formed LMDPE composite reinforced with agave fiber. The manuscript is readable, and the research results can provide a reference for industrial production. In my opinion, it can be accepted after following revisions. Although the authors reviewed some recent researches in the introduction part, which is good, the objective of their investigation is not clear enough. The authors should give a summary in each review paragraph and explain how these studies are relevant/differ to the current study.
Response: Thank you for the comment. The following information has been added to the introduction section (line 107):

One of the strengths of this work is the use of agave fibers (an agro-waste of tequila production in Mexico) for the processing and formulation of bilayer materials using a composite in the external layer and a polymer foam in the internal surface, in order to increase the sustainability by reducing the cost/weight ratio of the parts and the dependence of oil-based polymers. As reported above, these systems are commonly composed only by polyethylene and foamed polyethylene.

Additionally, throughout the introduction section (Lines 69-136), more information was added in order to state how this work is relevant/different from the cited literature.

Comment 2: Line 91, if the “rotomolded” is same as “rotational molding”, please uniform the appellation.
Response: Thank you for the comment. Both terms are commonly used in the literature to describe the processing technology. However, the term “rotomolding” was changed to “rotational molding” throughout the manuscript (lines 65, 153, 220 and 226). We would like to keep the term "rotomolded" because indicates that the pieces were produced by rotational molding.

Comment 3: Please give some explanations on the employment of agave fibers and why use the fiber size of between 100 mesh and 140 mesh.
Response: Thank you for your comments. The following information was added in section 2.2 Fiber preparation (line 151):

This particle size produced the best dispersion, distribution and mechanical performance, according to previous studies [16,23].

Comment 4: I have some confuse about the “Rotational Molding” processes, could you please show some detail about the mold, pictures or schema. What is the mold temperature and pressure during the molding process? How to composite the two layers?
Response: Thank you for the comment. A schematic (labeled as Figure 1) was added (line 172) in order to clarify on rotational molding process and to provide more details on bilayers production and test specimen cutting. In section 2.3 Rotational molding, all the processing parameters are described. In line 170, is informed that the mold has a vent conduct, for that reason, there is no pressure inside the mold, which is common in rotational molding. In this case, the venting was also used to load the second layer (as shown in Figure 1) and to place the thermocouple for internal air temperature monitoring, the mold temperature was not measured.

Comment 5: “3.2. Processing Cycle Time”, this part of the analysis is a bit superficial and redundant and is more suitable for presentation in the materials and methods section.
Response: Thanks for your suggestion. The following information was added to section 3.2 in order to point out the importance of this analysis (line 242):

The analysis of processing cycle time and internal air temperatures are key parameters due to the lack information available in the literature and their industrial significance. Rotational molding has longer cycle times, compared to other processing technologies, the raw materials are susceptible to suffer thermal degradation, particularly natural fibers. From this data, it was possible to confirm that the adequate processing conditions (e.g., oven temperature and heating times) were used, which guarantees the correct sintering process of the part, avoiding the thermal degradation of both fiber and matrix.

Comment 6: line 385, “FA”? Is it “AF”?
Response: Done.

Reviewer 2 Report

The authors present an interesting study on the obtention of bylayer composite materials by rotomolding. The use rotomolding processing is very interesting due to its costs and the possibility of manufacturing big pieces. The paper has been correctly written, the references are adequate and the state of the art has been exhaustive.

The materials and methods section describes all the operations, nonetheless I would like some details. The specimens were cut out from the rotomolded cylinder. I would like more details on this operation. From which zone of the cylinder do the authors cut out the specimens and what shape (dog bone, square…). Possibly a figure can be of help.

Finally, in line 4, please add the year for the waste generation predictions

Author Response

We would like to thank Reviewer 2 for the comments and suggestions that led us to improve the quality of the manuscript. Following the suggestions, the manuscript has been carefully edited and revised. All changes were typed in red color.

Comment 1: The materials and methods section describes all the operations, nonetheless I would like some details. The specimens were cut out from the rotomolded cylinder. I would like more details on this operation. From which zone of the cylinder do the authors cut out the specimens and what shape (dog bone, square…). Possibly a figure can be of help.
Response: Thank you for the comment. A schematic (labeled as Figure 1) was added (line 172) in order to provide more details on bilayers production and test specimens cutting. Also, the following information was added to line 191:

All the specimens for mechanical tests were cut from the rotomolded parts in the longitudinal direction of the cylinder (as shown in Figure 1), in order to avoid the curvature of the geometry of the piece. Finally, the specimens were tested at room temperature. All the samples were analyzed after one week to allow for stress relaxation before testing.

The geometry of flexural and impact specimens is already included in the corresponding sections. The geometry of tensile specimens was specified in line 206.

Comment 2: Finally, in line 4, please add the year for the waste generation predictions.
Response: Done.
